# Conditional Synthetic Data Generation for Robust Machine Learning Applications with Limited Pandemic Data

Hari Prasanna Das[1], Ryan Tran[1], Japjot Singh[1], Xiangyu Yue[1], Geoff Tison[2], Alberto
Sangiovanni-Vincentelli[1], and Costas J. Spanos[1]

[1]Department of Electrical Engineering and Computer Sciences, UC Berkeley,
[2]Division of Cardiology, University of California, San Francisco
{hpdas,bobotran,calzoom,xyyue,alberto,spanos}@berkeley.edu,geoff.tison@ucsf.edu

## ABSTRACT

**Background:** At the onset of a pandemic, such as COVID-19, data with proper labeling/attributes corresponding to the new disease might be unavailable or sparse. Machine Learning (ML) models trained with the available data, which is limited in quantity and poor in diversity, will often be biased and inaccurate. At the same time, ML algorithms designed to fight pandemics must have good performance and be developed in a time-sensitive manner. To tackle the challenges of limited data, and label scarcity in the available data, we propose generating conditional synthetic data, to be used alongside real data for developing robust ML models. **Methods:** We present a hybrid model consisting of a conditional generative flow and a classifier for conditional synthetic data generation. **Results:** We performed conditional synthetic generation for chest computed tomography (CT) scans corresponding to normal, COVID-19, and pneumonia afflicted patients. We show that our method significantly outperforms existing models both on qualitative and quantitative performance. As an example of downstream use of synthetic data, we show improvement in COVID-19 detection from CT scans with conditional synthetic data augmentation.

## CCS CONCEPTS

• **Computing methodologies → Neural networks**.

## KEYWORDS

COVID, Pandemic, Conditional Synthetic Data Generation

**ACM Reference Format:**
Hari Prasanna Das[1], Ryan Tran[1], Japjot Singh[1], Xiangyu Yue[1], Geoff Tison[2], Alberto Sangiovanni-Vincentelli[1], and Costas J. Spanos[1]. 2022. Conditional Synthetic Data Generation for Robust Machine Learning Applications with Limited Pandemic Data. In *epiDAMIK 2022: 5th epiDAMIK ACM SIGKDD International Workshop on Epidemiology meets Data Mining and Knowledge Discovery, August 15, 2022, Washington, DC, USA.* ACM, New York, NY, USA, 5 pages.

## 1 INTRODUCTION

The COVID-19 pandemic has created a public health crisis and continues to have a devastating impact on lives and healthcare systems worldwide. In the fight against this pandemic, a number of algorithms involving state-of-the-art machine learning techniques have been proposed. Data-based approaches have been used in a number of important tasks such as detection, mitigation, transmission modeling, decision on lockdown, reopening and related restrictions etc. For example, computer vision-based detection of COVID-19 from chest computed tomography (CT) images has been proposed as a supportive screening tool for COVID-19 [7], along with the primary diagnostic test of transcription polymerase chain reaction (RT-PCR). This is beneficial since obtaining definitive RT-PCR test results may take a lot of time in critical situations. Reinforcement learning based methods were also proposed to optimize mitigation policies that minimize the economic impact without overwhelming the hospital capacity [16].

The application of machine learning algorithms in healthcare depends upon ample availability of disease data along with their attributes/labels. At the beginning of a pandemic, data corresponding to the disease might be unavailable or sparse. Sparse data often have limited variation in several important factors relevant to disease detection such as age, underlying medical conditions etc. Class imbalance is another issue faced by machine learning algorithms when pandemic-disease related data is limited. For example, at the onset of COVID-19, the amount of CT scan images corresponding to COVID-19 were much less than those corresponding to other existing lung diseases (e.g. pneumonia). ML models fed with such class-imbalanced data could be biased and thus provide inaccurate results. Furthermore, the amount of data with proper labels among the available pandemic data might be minimal. This issue can arise because healthcare professionals and domain experts who can review and label the data are busy treating patients inflicted with the new disease, or also because of privacy concerns associated with medical data sharing.

Concurrently, after a new disease has been discovered, the healthcare ML tools must rapidly adapt to the new disease in order to assist medical professionals diagnose and treat affected individuals as quickly as possible. Rapid actions are also expected in design of policy interventions that are based on insights from pandemic data. Another issue in development of machine learning algorithms for emerging pandemics is privacy. Development of solutions to pandemics at the scale of COVID-19 require collaborative research which in turn presses the need for open-sourced healthcare data. But, even if healthcare organizations wish to release relevant data, they are often restricted in the amount of data to be released due to legal, privacy and other concerns.

In this paper, we present a novel conditional synthetic data-generation method to augment the available pandemic data of

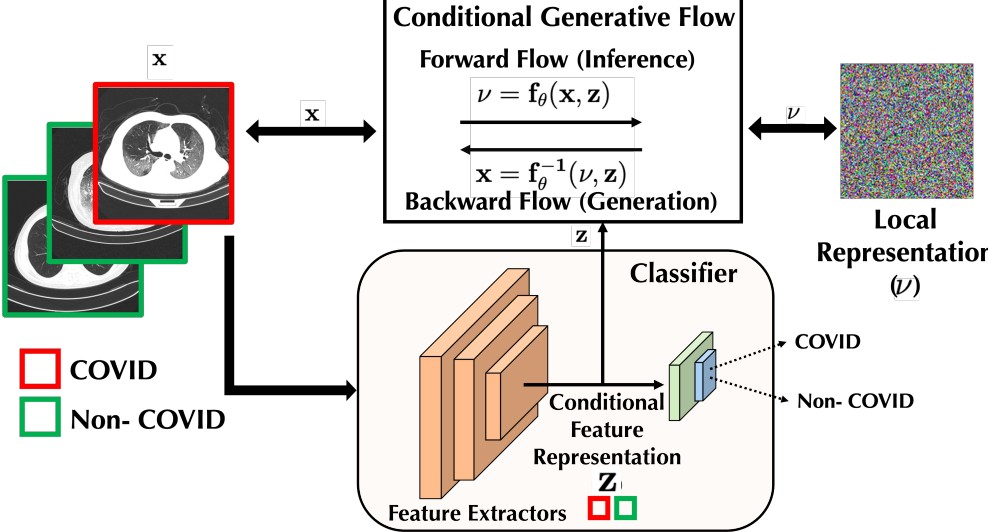

**Figure 1: Illustration of the proposed conditional synthetic generation. (Best viewed in color)**

interest. Our proposed method can also help organizations release synthetic versions of their actual data with similar behavior in a privacy-preserving manner. At the onset of a pandemic, when the availability of disease data is limited, our proposed model learns the distribution of available limited data and then generates conditional synthetic data that can be added to the existing data in order to improve the performance of machine learning algorithms. To tackle the challenge of label scarcity, we propose semi-supervised learning methods to leverage the small amount of labeled data and still generate qualitative synthetic samples. Our methods can enable healthcare ML tools to rapidly adapt to a pandemic.

We apply this method to generate conditional CT scan images corresponding to COVID cases, and conduct qualitative and quantitative tests to ensure that our model generates high-fidelity samples and is able to preserve the features corresponding to the condition (COVID/Non-COVID) in synthetic samples. As a downstream use of conditional synthetic data, we improve the performance of COVID-19 detectors based on CT scan data via synthetic data augmentation.

## 2 METHODOLOGY

We present a hybrid model consisting of a conditional generative flow and a classifier for conditional synthetic generation.

### 2.1 COVID and Non-COVID Classifier

Our model is characterized by the efficient decoupling of feature representations corresponding to the condition and the local noise. Suppose we have $N$ samples $\mathbf{X}$ with labels $Y$, with 2 possible classes, COVID/Non-COVID. We first train a classifier $C$ (consisting of a feature extractor network denoted by $g(\cdot)$, and a final fully-connected and softmax layer, denoted by $h(\cdot)$, i.e. $C(x) = h(g(x))$) to classify the input sample (which in our case are CT Scans) and associated labels as COVID and Non-COVID. Mathematically, this step solves

the following minimization with backpropagation:

$$\min_C \mathcal{L}_C(\mathbf{X}, Y) = -\mathbb{E}_{(x,y) \sim (\mathbf{X},Y)} \sum_{l=1}^{2} \left[ \mathbb{I}_{[l=y]} \log C(x) \right] \quad (1)$$

By virtue of the training process, the classifier learns to discard local information and preserve the features necessary for classification (conditional information) towards the downstream layers. Once the classifier is trained, we freeze its parameters, and use it to extract the conditional (COVID/Non-COVID) feature representation $z = g(x)$ (as a vector without spatial characteristics) at the output of the feature extractor network for input image $x$. The dimension of $z$ is chosen such that $\dim(z) << \dim(x)$.

### 2.2 Conditional Generative Flow

During the training phase for the flow model, the conditional feature representation $z$ is fed to the conditional generative flow. The flow model is trained using maximum-likelihood, transforming $x$ to its local representation $\nu$, i.e.

$$f_\theta(x, z) = \nu \sim \mathcal{N}(0, I) \quad (2)$$

with $\nu$ having the same dimension as $x$ by the inherent design of flow models. We use the method introduced by Ma et al. [19] to incorporate the conditional input $z$ in flow model. Coupling layers in affine flow models have scale ($s(\cdot)$) and shift ($b(\cdot)$) networks [2, 5], which are fed with inputs after splitting, and their outputs are concatenated before passing on to the next layer. We incorporate the conditional information $z$ in the scale and shift networks. Mathematically, (with $x$ as the input, $D$ as input dimension, $d$ as the split

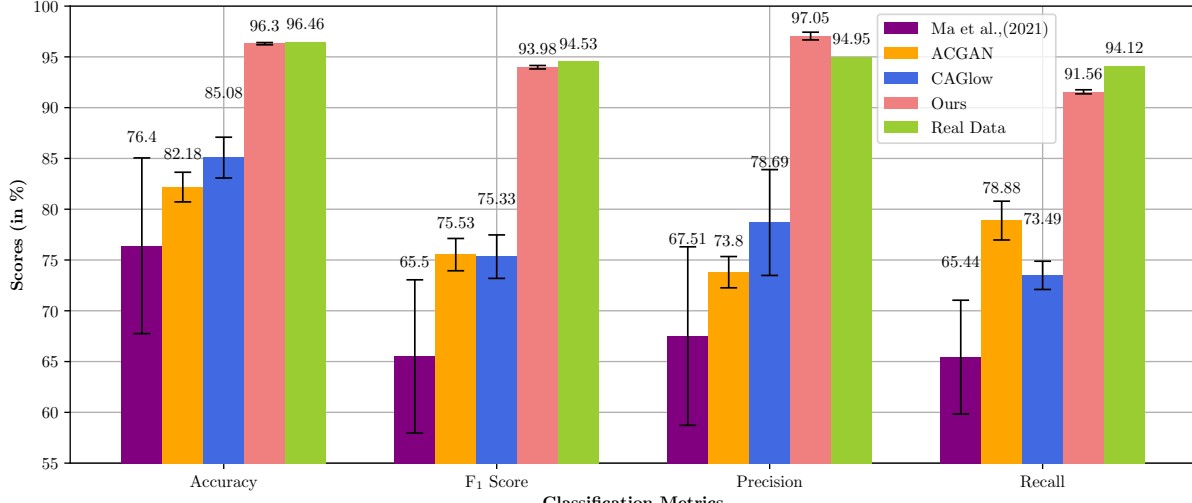

Figure 2: Classification metrics for classifiers trained on synthetic data generated by various models. The error bars indicate the variation in classifier performance when the synthetic datasets used to train them were generated multiple times with different seeds. Real data classifier does not involve multiple synthetic data generation, so its error bars are not included.

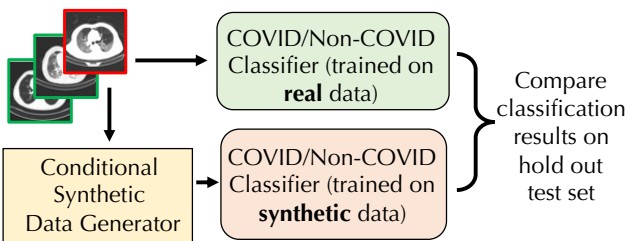

Figure 3: Illustration of quantitative testing procedure for conditional synthetic generation.

| Model | FID |
|---|---|
| Ma et al. [19] | 0.2504 |
| ACGAN | 0.0986 |
| CAGlow | 0.0483 |
| Ours | **0.0077** |

Table 1: FID scores (the lower the better).

size, and $y$ as output of the layer),

$$x_{1:d}, x_{d+1:D} = \text{split}(x)$$
$$y_{1:d} = x_{1:d}$$
$$y_{d+1:D} = s(x_{1:d}, z) \odot x_{d+1:D} + b(x_{1:d}, z)$$
$$y = \text{concat}(y_{1:d}, y_{d+1:D})$$

Since flow models are bijective mappings, the exact $x$ can be reconstructed by the inverse flow with $z$ and $v$ as inputs. During the generation phase, for an input sample $x$, we compute the conditional feature representation $z$. Keeping the conditional feature representation the same, we sample a new local representation $\tilde{v}$, and generate a conditional synthetic sample $\tilde{x}$, i.e.

$$\tilde{v} \in \mathcal{N}(0, I), \quad \tilde{x} = f_\theta^{-1}(\tilde{v}, z) \tag{3}$$

Here, $\tilde{x}$ has the same conditional (COVID/Non-COVID) features as $x$, but has a different local representation. An illustration of the proposed model is provided in Fig. 1.

## 3 EXPERIMENTS

**Data Collection:** We conduct experiments on chest CT scan data based on the COVIDx CT-1 dataset [7].The dataset consists of 45,758 images for healthy individuals, 36,856 images for individuals afflicted with common pneumonia, and 21,395 images for individuals with COVID-19.

**Pre-processing:** We combine the images in the Normal and Pneumonia classes into a single Non-COVID class. We use the train, validation, and test splits defined by the official annotation files. In addition to class labels, the annotations include bounding boxes for the lungs region in the whole CT scans image. We crop the images as per the bounding box and resize them to $64 \times 64$.

**Testing Procedure:** We performed both quantitative and qualitative testing for conditional synthetic data generation by our model. A test set is held out from the real dataset to be used for quantitative testing. We then compare the classification performance (COVID/Non-COVID) on this test set for a classifier trained on real data vs a classifier trained on the generated synthetic data. This testing procedure is illustrated in Fig. 3. Since the dataset is imbalanced, we report the precision, recall and macro-$F_1$ score (together referred to as classification metrics) along with the accuracy. For more information on the metrics, please refer to Hossin and Sulaiman [11]. Closeness of the classification metrics of classifiers

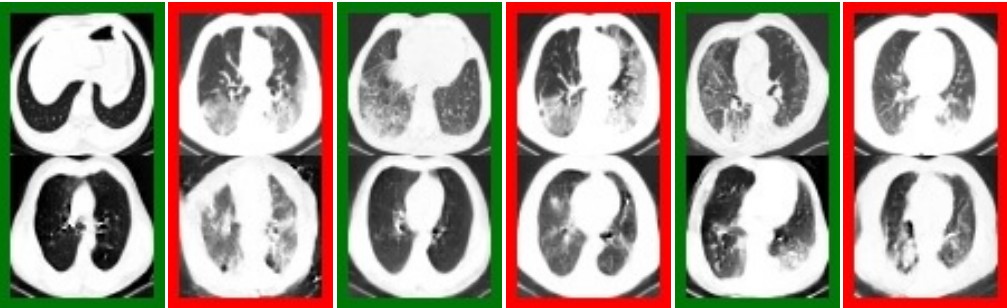

**Figure 4: Original and generated synthetic CT scan samples. The top row consists of original samples, and corresponding image in the bottom row is the synthetic sample obtained by preserving the original conditional feature representation, and varying the local noise. Image pairs with a red border: COVID samples, and a green border: Non-COVID samples.**

trained on synthetic and real data indicates an efficient design of the conditional synthetic generator. To evaluate the quality of generated samples, we report the Fréchet Inception Distance (FID) [9] for the synthetic samples. For FID calculation, we use the embeddings from our classifier trained using real data, in place of the official inception network [21], since the latter is not trained on CT scans.

## 4 RESULTS

The classification results for a classifier trained on the real data vs a classifier trained on purely conditional synthetic data, and tested on a hold-out set of real data, is given in Fig. 2. Across the existing methods for conditional synthetic generation, the classifier trained with synthetic data from our proposed model has the closest accuracy, $F_1$ score, precision and recall to that of the classifier trained on real data. This shows the capability of our method to generate synthetic samples with a distribution that closely matches the real conditional data distribution. The qualitative results (FID scores) for synthetic data generated by various models are tabulated in Table 1. The FID scores for our model is the lowest among all models, demonstrating that the quality of the generated samples closely matches the real ones.

It is worth noting that the accuracy/$F_1$ score of the classifier trained with synthetic data generated by Ma et al. [18] is much smaller as compared to those by other models, not to mention the classifier trained on real data. This can be justified from the fact that Ma et al. [18] relies on an unsupervised method of decoupling global and local information. But for conditional synthetic generation applications, such as the one presented in this paper, the model needs information on what the model designer/ domain experts consider as the conditional information (COVID/Non-COVID in our case). ACGAN and CAGlow have different generators, but both include an auxiliary supervision signal to conditionally guide the generation process. Hence, performance of classifiers trained on synthetic data generated by them are close. We encode the conditions using feature extractors to feed to the generator, leading to state-of-the-art results.

The original samples along with the synthetic samples generated by preserving original conditional feature representation and a different local noise for CT scans are shown in Fig. 4. The characteristic

features for COVID CT scans, i.e., ground-glass opacity are well preserved in the synthetic samples. The non-conditional local features, e.g. axial plane position are considered as local noise. Since original samples for normal and pneumonia cases are merged together to form the Non-COVID class, sometimes the corresponding synthetic image for a normal sample is a sample with pneumonia characteristics and vice-versa. This occurs since the conditional model learns to treat them as local information. The ability to decouple the feature representations for given conditions from other information in the data, as exhibited by our model, should be considered the strength of an effective conditional generative model.

## 5 RELATED WORK

In the field of healthcare, synthetic data generation has been proposed to expand the diversity and amount of the existing training data, often to improve the robustness of machine learning models. Ghorbani et al. [6] propose a generative adversarial network (GAN)-based synthetic data generator to improve the diversity and the amount of skin lesion images. Kohlberger et al. [15] synthesize pathology images for cancer with realistic out-of-focus characteristics to evaluate general pathology images for focus quality issues. Han et al. [8] propose synthetic generation to produce high-resolution artificial radiographs. In the space of combating COVID-19, Bannur et al. [1] propose a method of strengthening the COVID-19 forecasts from compartmental models by using short term predictions from a curve fitting approach as synthetic data. Similarly, Waheed et al. [22] and Jiang et al. [12] propose a conditional GAN-based generator for synthetic chest X-ray/CT scan data generation and augmentation for robust COVID-19 detection. Above works do not focus on the case where data with proper labels might be unavailable or sparsely available, whereas we tackle this challenge using a semi-supervised approach. We also show the robustness achieved using our model via experiments with several bootstrapping methods.

In the area of conditional generation, a hybrid flow and a GAN-based model have been proposed in CAGlow [17]. In general, GAN-based methods are known to be hard to train [20] and do not provide a latent embedding suitable for feature manipulations [13]. In contrast, we proposed a conditional generation method with efficient decoupling of the conditional information and local noise over an

embedding space, along with a flow based generator, which recently have proved efficient in synthetic data generation [3, 10].

Decoupling of global and local representation for synthetic generation has been proposed in Ma et al. [19], where the global information is decoupled using a Variational AutoEncoder (VAE) [14]. For conditional synthetic generation, it is necessary that the feature representations salient to the given conditions (COVID/Non-COVID) are decoupled from local noise, which is not guaranteed while extracting the same using a VAE. By employing a classifier network for the same, we ensure the relevant conditional information is not lost into the local noise.

## 6 DISCUSSIONS

We presented a novel conditional synthetic generative model aimed at multiplying the samples of interest at the onset of a pandemic. We conducted extensive experiments on chest CT scan dataset to show the efficacy of the proposed model, and improvements in COVID-19 detection performance achieved via synthetic data augmentation. Details on the method, experiments and results can be found in [4].

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
