# OpenReview forum: "Conditional Synthetic Data Generation for Robust Machine Learning Applications with Limited Pandemic Data"
_ACM.org/SIGKDD/2022/Workshop/epiDAMIK — KDD 2022 Workshop epiDAMIK Oral_

### Official Review · Reviewer_zK2L · 2022-06-18
**The method of conditionally generating synthetic data has direct applicability to epi domain, especially when there is a shortage of data. Overall, the method is sound, backed up with results.**

**Rating:** 3
**Confidence:** 3

**Review:**

Overall, the methodology and the results are sound. Here is a list of strengths, weaknesses, and suggestions of the paper

- Strength

The method performs better than the other synthetic data generation methods. Their method is directly applicable to the epi domain, especially when there's a shortage of data.

- Weakness

The results of using purely conditional synthetic data perform worse than using the real data (Figure 2). I suggest authors do an experiment by augmenting the real data with the synthetic data learned from their method, train the model, test it on the test set and show the performance gain.

- Suggestions
  - The discussion section is short and abruptly ends with an extended paper link to the AAAI paper. I suggest the authors summarize results that are not discussed in the paper.
  - I suggest authors follow the submission guideline to provide names and affliations.
  - I suggest authors stick with the formatting guidelines and either shorten (2-4 pages for short paper) or lengthen (6-8 pages for full regular research paper) the paper (currently at 5 pages).

---

### Official Review · Reviewer_odnp · 2022-06-24
**Conditional Synthetic Data Generation for Robust Machine Learning Applications with Limited Pandemic Data- Review**

**Rating:** 4
**Confidence:** 3

**Review:**

At the onset of any pandemic, limited data availability is a major concern for mitigating ML models trained on datasets to understand and prevent diseases. This calls for accurately generated synthetic data. This work proposes a hybrid model consisting of a classifier and a conditional generative flow model for conditional synthetic data generation. The model is used to generate synthetic CT images. The work also does a survey on the relevant literature and compares results with them.

Strong Points:
1. The work does a relevant literature survey and provides significant contributions to that domain with the current model.
2. Results are well presented and concise.
3. The model methodology is well explained.

Weak points:
1. The train: test split ratio should have been mentioned.
2. In figure 2, the precision of the classifier on generated data is more than that of the real data. This could mean that the generated data might be producing images that might not be as noisy as the real image (or it might be missing some characteristic of the original image which probably cannot be learned). This leads to the question of how do we understand whether the generated images are similar real images? Just using a classifier might not be enough. Probably a deep dive into some sort of metric would be a good addition. Something like the t-SNE visualization done in the TIMEGAN paper (Yoon et al., 2019) is what might do the validation of this.

---

### Official Review · Reviewer_aiwr · 2022-06-27
**Promising conditional synthetic data generation method, but only tested on one dataset**

**Rating:** 3
**Confidence:** 3

**Review:**

The authors propose a conditional synthetic data generation method that consists of a hybrid model: a conditional generative flow and a feature extractor network (part of a classifier) for conditional synthetic generation. The authors evaluate the proposed synthetic data generation technique on a publicly available dataset (chest CT scan), compare its performance to existing models, and show improvement over state-of-the-art.

Pros:

- The clarity of the article is appreciable, whether it is the introduction, figure 1, or the methodology.
- An extensive study has been conducted to highlight the performance of the proposed data augmentation model. The confidence intervals using multiple seeds in figure 2 are a great addition.

Major comments:

- The experiments have only been conducted on the chest CT scan dataset. Having the model tested on a second dataset would have been of great value (chest X-rays as an example, as it has been used in related studies).

Minor comments:

- The authors should include a baseline for the classification metrics. For the accuracy, is it around 44% (based on the size of the dataset and the methodology)?
- The individuals afflicted with pneumonia classed as Non-COVID is a choice made by the authors. Nonetheless, it would have been interesting to train models with 3 labels, or just to train models on the Normal and COVID individuals and examine the differences in performance.
- As highlighted by the authors, “[…] synthetic data (that) can be added to the existing data in order to improve the performance of machine learning algorithms”. The accuracy of the model when combined with both real and synthetic data may support the aforementioned statement.
- Please provide a GitHub repository of the codes to allow replication of the analysis.